# Robot-Assisted Total Thyroidectomy with or without Robot-Assisted Neck Dissection in Pediatric Patients with Differentiated Thyroid Cancer

**DOI:** 10.3390/jcm11123320

**Published:** 2022-06-09

**Authors:** Dahee Kim, Nam Suk Sim, Dachan Kim, Eun Chang Choi, Jae Won Chang, Yoon Woo Koh

**Affiliations:** 1Department of Otorhinolaryngology, Severance Hospital, Yonsei University Health System, Yonsei University College of Medicine, Seoul 03722, Korea; dhk@yuhs.ac (D.K.); artmasim@yuhs.ac (N.S.S.); dckim20@yuhs.ac (D.K.); eunchangmd@yuhs.ac (E.C.C.); 2Department of Otorhinolaryngology, Chungnam National University, Daejeon 34134, Korea

**Keywords:** robot-assisted total thyroidectomy, robot-assisted modified radical neck dissection, transaxillary and retroauricular approach, retroauricular approach, pediatric thyroid cancer

## Abstract

Pediatric thyroid cancer more frequently develops cervical node metastasis than adult thyroid cancer, even in differentiated thyroid carcinoma (DTC). Thus, cervical neck dissection often needs to be performed simultaneously with thyroidectomy in pediatric patients. Herein, we describe our experience with robot-assisted total thyroidectomy with/without robot-assisted neck dissection in pediatric patients compared with the conventional operated group. A total of 30 pediatric patients who underwent thyroidectomy for DTC between July 2011 and December 2019 were retrospectively reviewed. Among them, 22 underwent robot-assisted operation, whereas 8 underwent conventional operation. There was no statistical difference in the mean operation times, blood loss, drainage amounts, and hospital stay length between the robot-assisted and conventional operation groups; however, the operation time was less in the retroauricular approach subgroup (robot-assisted operation group) with better satisfaction on cosmesis. No postoperative complications, such as seromas, hemorrhages, or hematomas were observed. Our experience suggested that robot-assisted thyroidectomy with or without neck dissection through the retroauricular approach is a feasible and safe alternative treatment, producing outstanding esthetic results compared to the conventional approach, especially in pediatric patients with DTC.

## 1. Introduction

Although thyroid cancer rarely occurs in childhood, it accounts for approximately 0.7% of all cancers and 1–3% of all pediatric malignancies; it is the most common pediatric endocrine malignancy having an increasing incidence [1]. Accumulating evidence suggests that pediatric papillary thyroid cancer (PTC), the most common pathology among pediatric thyroid carcinomas (papillary variant, 60%; follicular variant, 23%), has a different clinical presentation and course from that of adult PTC [2,3]. Particularly, PTC tends to present with a larger primary tumor, wider extrathyroidal extension, and higher incidences of neck lymph node and distant metastases in pediatric than in adult patients [4]. However, even in advanced stages or with distant metastasis, the prognosis is excellent for pediatric patients, with a low mortality rate. Furthermore, the 30-year survival rate was reported to be 90–99% in patients with pediatric differentiated thyroid malignancies [5].

Thyroid cancer management in pediatric patients, including surgical principles, is largely extrapolated from evidence obtained from adult patients [6]. However, the optimal surgical extent remains controversial. A generally accepted fact is that total thyroidectomy with central compartment lymph node dissection has been known to reduce recurrence rate but carries a relatively higher risk of permanent hypoparathyroidism and vocal cord paralysis due to recurrent laryngeal nerve (RLN) injury in pediatric patients than in adult patients [5]. Additionally, several studies have described a substantially higher risk of complications after thyroidectomy in pediatric patients than in adult patients [7]. Therefore, less radical surgery (lobectomy) has been advocated by some authors, emphasizing the lower risks of surgical and postoperative morbidity [8]. However, we postulated that if better oncological safety with lower morbidity and better cosmesis can be achieved by using robot-assisted surgical techniques, which have been actively investigated and applied, the pediatric thyroidectomy surgical extent may be clarified.

Since Gagner et al. first described endoscopic parathyroidectomy [9], different types of minimally invasive endoscopic or video-assisted approaches have been developed for thyroid surgery [10]. However, these techniques also have several limitations [11]. Recent innovation in robotic engineering and surgical systems have allowed the development and active application of various remote access techniques for thyroidectomy, while overcoming the limitations of previously developed minimal invasive surgeries. Particularly, the most widely acknowledged robot-assisted thyroidectomy technique uses a gasless transaxillary approach, which has been introduced, developed, and well documented by Chung et al. [12]. Particularly, it is associated with a shorter learning curve than endoscopic surgery and has caused less musculoskeletal discomfort compared to open or endoscopic techniques. Regarding function, it has yielded outstanding cosmesis, as well as reduced neck pain, sensory changes, and voice or swallowing discomfort after surgery [13]. Conversely, cervical lymph node dissection with thyroidectomy in the transaxillary approach has an inherent limitation in upper lateral neck dissection (cervical neck levels I, IIa, and Va). In this regard, over the past decade, researchers have developed a unilateral retroauricular approach via the transaxillary and retroauricular approach in robot-assisted total thyroidectomy with robot-assisted neck dissection and have also reported the feasibility and safety of these techniques [14,15]. Currently, with far more advanced robotic systems, the retroauricular approach is sufficient to expose both the thyroid and neck at all levels.

The aim of this study was to compare the thyroidectomy techniques and outcomes of robot-assisted and conventional surgery in pediatric populations and to introduce our surgical experience after successfully attempting robot-assisted total thyroidectomy with or without neck dissection in these patients. To the best of our knowledge, this is the first report on pediatric robot-assisted thyroidectomy in the international literature.

## 2. Materials and Methods

### 2.1. Patients

A total of 30 patients (23 females and 7 males) aged ≤20 years (mean age, 16.6 ± 2.7 years) who underwent conventional or robot-assisted thyroidectomy with or without neck dissection at the Yonsei Head and Neck Cancer Center and Chungnam National University between July 2011 and December 2019 were enrolled in the present study. The inclusion criteria were as follows: (i) pediatric patients, defined as those aged 20 or less; (ii) patients with histopathologically proven malignant differentiated thyroid gland carcinomas with or without cervical lymph node metastasis on preoperative imaging studies who were indicated for surgery; and (iii) patients with no previous history of thyroid carcinoma treatment. The exclusion criteria were as follows: (i) patients with recurrent thyroid tumors; (ii) patients with thyroid carcinomas showing evident adjacent structure invasion or extensive extrathyroid parenchymal spread; (iii) patients with cervical metastasis and clinically evident extracapsular spread; and (iv) patients with a history of any kind of neck surgery. To assess disease extent, preoperative evaluations were performed using neck ultrasonography with fine-needle aspiration, neck computed tomography or magnetic resonance imaging, and positron emission tomography-computed tomography. All the patients and their families or guardians were given full information on the possible treatment options for their thyroid cancer, including open transcervical approach and robot-assisted surgery via each approach, with the advantages and disadvantages of each treatment choice. Written informed consent was obtained from all patients or their guardians before surgery. The robot-assisted operations costed more than the conventional operations by approximately USD 5000; however, they were covered by the Korean personal health insurance. General clinical characteristics of the patients are shown in Table 1.

### 2.2. Surgical Technique

All operations were performed by two experienced head and neck surgeons (Y.W.K and J.W.C). Until 2013, we used the transaxillary approach for robot-assisted thyroidectomy without lateral neck dissection and the transaxillary–retroauricular approach for the robot-assisted thyroidectomy with modified neck dissection, as previously described by the Yonsei Medical Center. Since 2013, however, we have only used the retroauricular skin incision approach for robot-assisted thyroidectomy with or without lateral neck dissection, which was modified from the approach suggested by Terris et al. [16] for robot-assisted facelift hemithyroidectomy and the robot-assisted neck dissection [17], which was first reported by our group. Detailed procedures and overall concepts for the transaxillary–retroauricular and retroauricular approaches have been described previously and are summarized here. The overall sequence of the operation was as follows: retroauricular incision, working space creation, upper neck dissection under gross macroscopy (additional axillary incision for the transaxillary–retroauricular approach), robot-assisted neck dissection of the lower neck, and robot-assisted thyroidectomy with central compartment lymph node dissection after creation of working space for thyroidectomy.

#### 2.2.1. Robot-Assisted Modified Radical Neck Dissection (Levels II, III, IV, and V)

The patient was placed in the supine position with the neck slightly extended. Both axillary and retroauricular skin incisions were made in exactly the same way as that in adult patients described previously [14,15]. In brief, a retroauricular incision was made around the postauricular sulcus, elevating the hairline and subplatysmal skin flap to obtain a sufficient working space. The elevated skin flap was then retracted and maintained with a self-retaining retractor (Sangdosa Inc., Seoul, Korea), which was modified from Chung’s retractor. Afterward, the posterior belly of the digastric muscle was exposed by dissecting the lower margin of the submandibular gland. The spinal accessory nerve (SAN) was then identified at the posterior border of the sternocleidomastoid muscle (SCM), subsequently skeletonizing its entire course. Following this, level II, upper level III, and level Va tissues were dissected under direct macroscopy, with the SCM retracted laterally.

For the retroauricular approach, a 30° dual-channel endoscope was placed facedown on the central arm of the da Vinci surgical robot (Intuitive Surgical, Sunnyvale, CA) and inserted in the midline of the retroauricular skin incision. Each side of the endoscope was equipped with Harmonic curved shears (Ethicon, Johnson & Johnson Company, New Brunswick, NJ, USA), and 5 mm Maryland forceps. After setting up the endoscope, the robot-assisted neck dissection of lower levels III, IV, and Vb was performed [14].

For the transaxillary–retroauricular approach, a 7 cm additional incision was made at the anterior border of the axillary fossa, followed by skin flap elevation above the pectoralis major, which was elevated continuously up to the previously dissected level of upper neck and medially to a level higher than the contralateral thyroid gland. After applying the self-retaining retractor (Chung’s retractor), the robotic system was set up as previously described by our group, and robot-assisted neck dissection was performed for lower levels III, IV, and Vb via the transaxillary incision [15].

#### 2.2.2. Robot-Assisted Total Thyroidectomy with Central Compartment Node Dissection

After collecting the tissue specimens obtained using modified radical neck dissection, robot-assisted total thyroidectomy with central compartment node dissection was performed via either of the approaches described previously [14,15]. Briefly, after division and identification of superior thyroid vessels and ligation with Harmonic curved shears, the upper pole of the thyroid gland was released. Next, the thyroid gland was dissected from the trachea toward the lower pole, carefully dissecting it from the surrounding tissues for RLN identification. During this process, the parathyroid glands were identified at the posterior surface of the thyroid gland and were dissected off the gland posteriorly. Then, dissection continued downward toward the ipsilateral central compartment node dissection. After completing thyroid isthmusectomy, the rest of the thyroid lobe was dissected away from the trachea, and the connecting Berry’s ligament and the specimen were removed. Initiated by dissecting the remaining thyroid isthmus, contralateral thyroid gland dissection was performed in a medial-to-lateral direction with subcapsular dissection, while preserving the parathyroid glands and the RLN. Central compartment lymph node dissection was then performed while taking care not to damage the nerve. For contralateral thyroidectomy, when necessary, the instrument arm holding the Maryland dissector forceps was changed to ProGrasp (Intuitive Surgical) forceps to retract the gland. After specimen removal, the operation field was irrigated, and a closed suction drain was positioned through the same incision. After repairing the subcutaneous layer, the axillary incision was closed with a vertical mattress suture, and the retroauricular incision was closed with a topical skin adhesive.

## 3. Results

Among 30 patients, 22 underwent the robot-assisted approach, whereas 8 underwent the conventional approach. To classify by scope of surgery, 12 underwent robot-assisted total thyroidectomy with modified neck dissection (3 via the transaxillary–retroauricular approach, 6 via the retroauricular approach, and 3 via conventional approach); 7 underwent total thyroidectomy without neck dissection (2 via the transaxillary approach, 1 via the retroauricular approach, and 4 via the conventional approach); and the remaining 11 patients underwent hemithyroidectomy (2 via transaxillary approach, 8 via retroauricular approach, and 1 via conventional approach) (Figure 1). For all robot-assisted thyroidectomies with/without robot-assisted neck dissection via the unilateral transaxillary/transaxillary–retroauricular or retroauricular approach, the operation was successfully completed with no intraoperative complications or procedure conversions. Furthermore, the working space created with each approach was sufficient, and robotic tool manipulations through this space were technically feasible and safe with no collisions(Figure 2).

Pathologically, the tumors were largely identified as conventional papillary thyroid carcinomas (92%), with one patient of adenomatous hyperplasia and one patient of follicular carcinoma. Tumor size did not differ between groups, with tumors measuring up to 3.5 cm in each group, ranging from 0.5 cm to 3.9 cm in maximal diameter. Extrathyroidal extension rate was 50% (14 out of 28), which was higher than that reported in previous studies on adult thyroid cancer, and the mean total number of lymph nodes retrieved was 7.6 ± 3.7 via central compartment node dissection and 39.2 ± 24.0 via neck dissection. Furthermore, lymph node retrieval rates were higher via the retroauricular and conventional approaches than those via the transaxillary approach; however, a sufficient number of lymph nodes were retrieved via all three approaches of modified neck dissection.

Total operation time, defined as the time from initial incision to specimen removal, was 377.5 ± 98.8 min (mean ± SD) for total thyroidectomy with neck dissection, 204.5 ± 52.9 min for total thyroidectomy without neck dissection, and 167.3 ± 3.2 min for hemithyroidectomy (Table 2). These durations included times for skin flap elevation and neck dissection under gross macroscopy (84.6 ± 5.9 min); setting up the robotic system for robot-assisted neck dissection (6.2 ± 1.6 min); console time using the robotic system for robot-assisted neck dissection (100.8 ± 25.8 min); flap elevation for thyroidectomy (11.4 ± 1.9 min); docking the robotic arms for ipsilateral thyroidectomy (6.1 ± 2.2 min); ipsilateral thyroidectomy (57.7 ± 9.8 min); docking the robotic arms for contralateral thyroidectomy (4.3 ±0.9 min); and the console time for contralateral thyroidectomy (56.4 ± 12.2 min) in robot-assisted approach groups. Robot-assisted hemithyroidectomy required longer operation time in both transaxillary and retroauricular approaches, with statistical significance (robot-assisted hemithyroidectomy (167.3 ± 33.2 min) versus conventional hemithyroidectomy (92.0 ± 0.0 min) (*p*-value 1.46 × 10^−7^)), possibly due to the time gap required to create the working space in the robot-assisted group. Robot-assisted total thyroidectomy also required longer operation time, but no statistical significance was noted (robot-assisted total thyroidectomy (204.5 ± 52.9), than conventional total thyroidectomy (168.3 ± 10.2 min) (*p*-value = 0.13)). However, if we perform a subgroup analysis based on the two approaches within the total thyroidectomy group, the transaxillary and conventional approaches still showed no statistically significant difference, but retroauricular approach showed a dramatically reduced operation time (robot-assisted retroauricular approach—total thyroidectomy 120.0 ± 0.0 min). For total thyroidectomy with neck dissection patients, overall operation time was comparable between the robot-assisted group and the conventional group (total thyroidectomy with neck dissection—robot-assisted (377.5 ± 98.8) vs. total thyroidectomy with neck dissection—conventional (350.0 ± 14.1 min) (*p*-value = 0.26)) However, the transaxillary–retroauricular approach required a longer operation time than the conventional approach since both transaxillary and retroauricular working spaces had to be utilized for full exposure, whereas the retroauricular approach alone showed a comparable, and even shorter operating time than the conventional approach, which may be affected by updates of the latest robotic systems (total thyroidectomy with neck dissection—retroauricular (335.0 ± 57.5 min); total thyroidectomy with neck dissection—transaxillary/transaxillary–retroauricular (534 ± 5.6 min); total thyroidectomy with neck dissection—conventional (377.5 ± 98.8 min)) (Table 3). Additionally, the estimated average blood loss during the course of the operation was 22.7 ± 13.1 mL with no statistically significant differences between the robot-assisted and conventional groups (*p* = 0.698).

None of the patients had postoperative seromas, hemorrhages, or hematomas. Instead, most of the patients had postoperative vocal cord palsy due to RLN traction injury, except for one patient in the total thyroidectomy with neck dissection group via the transaxillary approach. Meanwhile, four patients developed transient hypoparathyroidism with a tingling sensation but recovered within 3 months postoperatively, without the need for any permanent medical supplementation. Calcium and vitamin D supplements were prescribed until hypoparathyroidism was resolved.

The mean drainage amount and hospital stay duration showed no statistical difference between the robot-assisted and conventional operation, according to each operation scope. Average drainage amounts were 37.9 ± 32.2 mL in the hemithyroidectomy group (robot-assisted (38.7 ± 31.3) vs. conventional (34.2 ± 0.0), *p* = 0.55), 137.7 ± 119.0 mL in the total thyroidectomy group (robot-assisted (135.4 ± 111.6) vs. conventional (108.4 ± 36.4), *p* = 0.07), and 310.8 ± 163.8 mL in the total thyroidectomy with neck dissection group (robot-assisted (298.6 ± 154.1) vs. conventional (229.0 ± 79.1), *p* = 0.05). In particular, the wound drainage catheter was removed 5.6 ± 0.8 days after the operation in the total thyroidectomy with neck dissection group, and the patients were discharged on the next day. All 11 patients who underwent robot-assisted total thyroidectomy with robot-assisted neck dissection received high-dose (100–150 mCi) radioiodine ablation (RAI) therapy, while 3 of the 6 patients who underwent robot-assisted total thyroidectomy without robot-assisted neck dissection received low-dose (30 mCi) RAI after the operation. The mean follow-up period for all groups was 73 months (range, 17–151 months). During regular (after RAI and most recent) follow-up examinations, no evidence of recurrence or distant metastasis was found (non-suppressed thyroglobulin level, <0.1–0.8 ng/mL; antithyroglobulin antibody level, <10.0–72.9 IU/mL) in all patients. Furthermore, patients operated with retroauricular and transaxillary approach were extremely satisfied with their cosmetic results after the operation compared to the conventional operation group (average visual analogue score 1–10; robot-assisted 9.2 [transaxillary 9.5, retroauricular 9.1] vs. conventional 7.7).

## 4. Discussion

Pediatric thyroid carcinoma is rare, but with its increasing prevalence, thyroid surgery is now being performed in a growing number of pediatric patients [3,4,5]. Furthermore, pediatric thyroid cancer is characterized by high incidences of locoregional aggressiveness, multifocality, lymph node metastasis, and distant metastasis at the time of diagnosis [4]. Therefore, aggressive surgical approaches such as total thyroidectomy and central compartment node dissection combined with modified radical neck dissection should be performed more frequently in pediatric patients than in adult patients [4,18,19]. Additionally, a thyroidectomy scar constitutes a special concern in pediatric thyroid cancer patients, especially due to its location, patients’ young age, female predominance, and emotional vulnerability of adolescents [20,21]. Furthermore, if thyroidectomy is combined with lateral neck dissection, which is performed in 35–83% patients [22], the skin incision is required to be extended quite laterally (extended large collar incision), resulting in a disfiguring prominent scar on the pediatric patient’s neck [23]. In this regard, remote approaches aiming to minimize scar visibility for thyroid surgery have benefits particularly in pediatric patients than in adult patients. Therefore, in the present study, we demonstrated a novel technique for robot-assisted total thyroidectomy with or without robot-assisted modified radical neck dissection via the unilateral retroauricular approach alone in pediatric patients with differentiated thyroid cancer.

Recent advances in robotic technologies have enabled remote assessment, which is superior in terms of cosmesis and precision [24]. Chung et al., in particular, reported their first experience with robot-assisted thyroidectomy [10] and robot-assisted neck dissection [25] using the gasless transaxillary approach for thyroid cancer management. We adopted and modified their original approach by adding retroauricular incision to the axillary incision (hence called transaxillary–retroauricular approach) to overcome the difficulty in tissue visualization at levels IIa and Va in patients with a prominent clavicle [26]. Additionally, we recently reported a comparative study between our modified and conventional approaches in patients with thyroid cancer having cervical lymph node metastasis [15]. Furthermore, after gaining substantial experience using the transaxillary–retroauricular or retroauricular approach for head and neck or thyroid cancers, we reached the conclusion that the unilateral retroauricular incision was only sufficient for robot-assisted modified radical neck dissection and total thyroidectomy with central compartment node dissection. Omitting the axillary incision allowed for shorter operation times and achieved better results in terms of cosmesis and invasiveness [14]. In fact, Terris et al. suggested that the modified facelift approach required a 38% lesser dissection extent [27]. Additionally, from a surgeon’s point of view, the retroauricular approach would be more preferable to the transaxillary approach due to an increased familiarity of the anatomy, as well as reduced complications since the spinal accessory nerve, RLN, and central neck lymph nodes are easier to address with this approach [28]. Since the postoperative scar was hidden behind the hairline, the pediatric patients and their families were satisfied with the cosmetic results, similar to those of adult patients (Figure 3) [14].

In terms of perioperative complications, hypocalcemia is a common complication of total thyroidectomy, leading to patient discomfort, prolonged admission length, and increased medical costs [5]. Although the reported data on pediatric series are scarce, temporary hypocalcemia rate in pediatric patients after thyroidectomy via the transcervical approach has been reported to be between 7% and 52% [29]. In our study, four patients (14%) experienced temporary hypocalcemia that spontaneously resolved within 3 months, which may be caused by extensive dissection due to the extrathyroidal extension and infiltrative tumor margin observed in all postoperative hypocalcemia patients. Additionally, a study reported that the parathyroid gland was incidentally removed by the surgeon in 19% of the pediatric patients who received total thyroidectomy via the transcervical approach [29], which was a considerably better rate than those in several adult series reporting a 30% incidental parathyroid gland resection incidence rate [30]. In our patient series, we did not experience incidental parathyroid gland resection. Only one of our patients had a transient RLN injury, whereas in a large series of pediatric thyroidectomy patients, the reported incidence was approximately 6% [31]. This relatively low postoperative complication rate may be explained by the benefits of the robotic surgical system, such as having a three-dimensional magnified vision, stable operation view, computer-assisted motion scaling, and tremor elimination, all of which facilitate easier anatomical identification and a more meticulous dissection [32]. Interestingly, during the operation, we noticed that even in the patient with a huge and aggressive primary disease or lymph node metastasis, the tissues could be dissected more easily than those of adult patients. Although the exact reason is unknown, this could be used as a basis for applying robotic surgery in pediatric advanced thyroid cancer patients with large lateral neck metastases.

Operation times in our series were comparable with those in adult patients in our recently reported study [14], but a longer operation time and a relatively wider flap elevation due to the approach were the main disadvantages of this operative procedure [14,15]. However, we already demonstrated a significant decrease in mean operation time and higher lymph node retrieval rates with the accumulation of experience in performing robot-assisted neck dissection for head and neck cancers [33]. Similarly, the operation time in our patient series resulted in a lower learning curve and considering the most updated version of the da Vinci surgical system, the single-port system, demonstrated even smaller and more refined instruments, making it is easy to use in small pediatric patients.

Since the advantages of robot-assisted surgery included better cosmesis, higher magnified vision for precision surgery, and comparable surgical and oncological outcomes, we believe that robot-assisted thyroidectomy and neck dissection in pediatric populations may be feasible. In particular, regarding multifocality with high rates of lymph node metastasis and extrathyroidal extension in pediatric patients with high concern for social stigmatization, delicate surgery utilizing robotic systems may be highly favorable. Both retroauricular and transaxillary approaches for robot-assisted thyroidectomy can be equally effective in terms of operation time, hospital stay duration, complications, and surgery completeness. Furthermore, the retroauricular approach has several advantages in patients with lateral neck metastasis in terms of surgical procedure and perioperative outcomes.

The relatively small number of patients and the short follow-up period were the limitations of this study. Additionally, because the cost-effectiveness would differ among countries, the same approach might have limitations in the choice of options. However, this is the first clinical report of successful robot-assisted total thyroidectomy and robot-assisted neck dissection via unilateral single retroauricular incision in pediatric patients. To address these limitations, we are currently conducting a prospective study on long-term outcomes in relation to oncological safety and functional results in an expanded population.

## 5. Conclusions

Robot-assisted total thyroidectomy with or without robot-assisted neck dissection was technically feasible and effective in thyroid cancer treatment in our pediatric patients. Notably, thyroid cancer accompanied with lateral neck metastases was more frequent in pediatric patients than in adult patients. Thus, for pediatric patients, robot-assisted total thyroidectomy with robot-assisted neck dissection via the single retroauricular approach can be an alternative to conventional transcervical total thyroidectomy with neck dissection in terms of perioperative outcomes. However, functional and oncological outcomes of the procedure should be verified in a larger number of patients with longer follow-up periods.

## Figures and Tables

**Figure 1 jcm-11-03320-f001:**
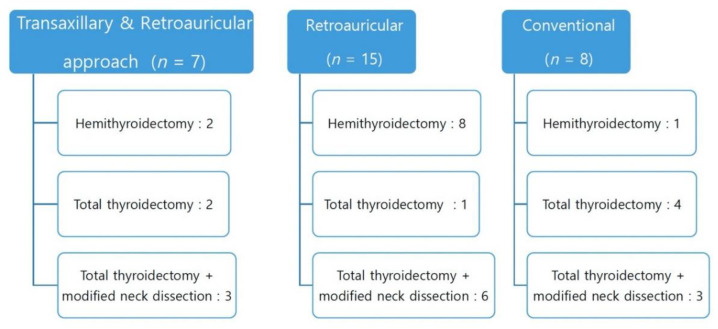
Schematic graph of types of operation performed and approaches undertaken for the study participants.

**Figure 2 jcm-11-03320-f002:**
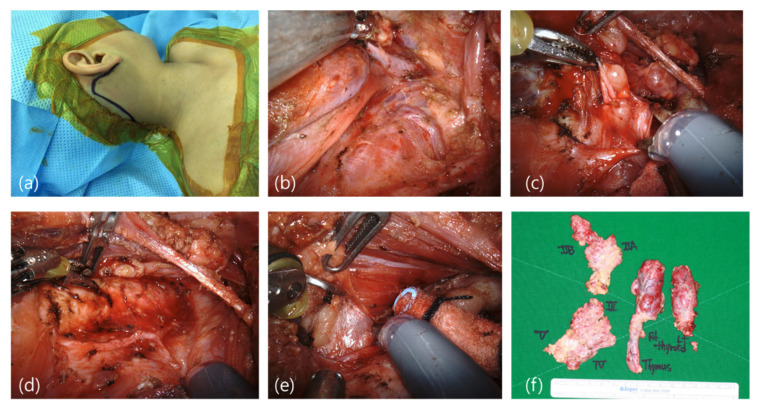
Robot-assisted neck dissection via a retroauricular port of patient RA13. (**a**) Incision and patient positioning (**b**) Level IV dissection. (**c**) Superior parathyroid exposed. (**d**) Surgical field after thyroidectomy. (**e**) Contralateral thyroidectomy view. (**f**) Surgical specimen retrieved after the completion of robot-assisted total thyroidectomy with robot-assisted modified radical neck dissection including levels II to V.

**Figure 3 jcm-11-03320-f003:**
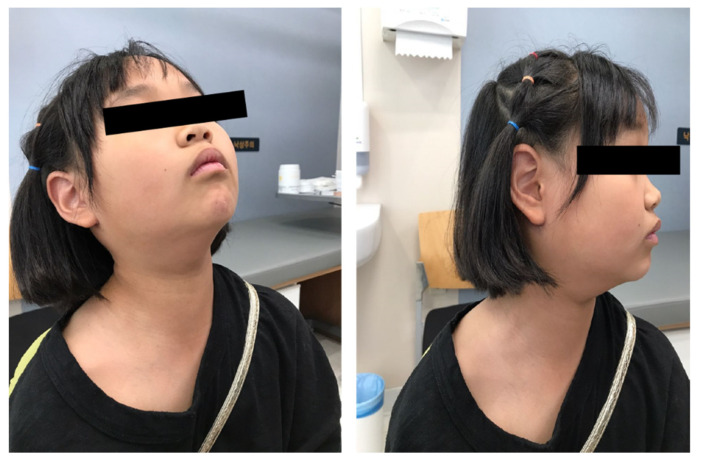
Postoperative photograph obtained 1 year after robot-assisted total thyroidectomy with robot-assisted modified radical neck dissection via the retroauricular approach.

**Table 1 jcm-11-03320-t001:** Clinical characteristics of the patients.

Patients	Gender/ Age (Years)	BMI (kg/m^2^)	Side ^a^	Operation	CCND	Pathology	Tumor Size	ETE	CND	MRND	ECS	Follow-Up (Months)
TA1	F/19	23.57	L	HT	+	AH	3.5	−	0/1			115
TA2	M/17	18.7	R	HT	−	PC	2	−	−			104
TA3	F/16	20.66	B	TT	−	PC	1.8	−	−			112
TA4	M/17	19.67	B	TT	+	PC	0.9	−	0/2			107
TA5	F/15	15.56	B	TT c MND	+	PC	2.7	+	3/6	9/32		104
TA6	M/14	20.27	R	TT c MND	+	PC	1.7	+	0/5	11/42	+	98
TA7	F/19	18.2	B	TT c MND	+	PC	3.5	+	0/1	22/64		97
RA1	F/13	18.29	L	HT	+	PC	2	+	0/7			79
RA2	F/16	27.85	R	HT	−	PC	1.2	−	−			57
RA3	M/15	24.44	R	HT	−	PC	3.5	−	−			93
RA4	F/17	23	L	HT	+	PC	3.5	−	0/5			76
RA5	F/18	20.74	L	HT	−	PC	0.9	−	−			50
RA6	F/18	14.68	R	HT	+	PC	3.3	+	2/5		+	27
RA7	M/18	18.69	R	HT	+	PC	0.5	+	1/9			26
RA8	F/18	24.22	R	HT	_	FC	1.2	−	−			107
RA9	F/17	22.99	R	TT	+	PC	1	−	3/9			92
RA10	F/18	18.34	L	TT c MND	+	PC	0.8	−	1/6	9/43		94
RA11	F/15	22.76	R	TT c MND	+	PC	1.9	+	0/14	3/65		75
RA12	F/17	28.93	R	TT c MND	+	PC	4	+	2/8	13/67		45
RA13	F/9	17.09	R	TT c MND	+	PC	1.5	+	1/5	8/92	+	41
RA14	M/13	19.01	R	TT c MND	+	PC	5	+	3/5	6/49		17
RA15	F/18	26.25	R	TT c MND	+	PC	1.1	−	5/10	8/65		17
Con1	F/16	25.9	R	HT	+	PC	2.5	+	7/7			72
Con2	F/20	20.7	R	TT	−	PC		−	−			143
Con3	F/16	18.03	L	TT	+	PC	0.8	+	0/2			104
Con4	F/16	21.65	B	TT	+	PC	1.3	+	2/4			101
Con5	F/10	16.94	L	TT	+	PC	0.6	−	0/4		−	
Con6	F/17	25.9	B	Completion thyroidectomy c MND	−	PC	3.9	−	0/1	10/52		72
Con7	F/14	28.44	L	TT c MND	+	PC	1.8	+	15/18	35/58	+	151
Con8	M/11	20.98	B	TT c MND	+	PC	4.1	+	7/11	19/89	+	30

Abbreviations: BMI, body mass index; CND, central compartment neck dissection; TA, transaxillary approach; TARA, transaxillary and retroauricular approach; HT, hemi thyroidectomy; TT, total thyroidectomy; RA, retroauricular approach; MND, modified radical neck dissection; F, female; M, male. AH. Adenomatous hyperplasia; PC, papillary carcinoma; FC, follicular carcinoma; ECS, extracapsular spread. ^a^ Side refers to the site of the main lesion.

**Table 2 jcm-11-03320-t002:** Operation time.

Approach	RA	TA/TARA	Robot-Assisted	Conventional	*p*-Value
Hemithyroidectomy	173.0 ± 32.0	119.5 ± 0.5	167.3 ± 33.2	92.0 ± 0.0	<0.001 *
Total Thyroidectomy	120.0 ± 0.0	256.0 ± 35.4	204.5 ± 52.9	168.3 ± 10.2	0.13
Total Thyroidectomy c Modified Neck Dissection	335.0 ± 57.5	534 ± 5.6	377.5 ± 98.8	350.0 ± 14.1	0.26

Abbreviations: TA, transaxillary approach; TARA, transaxillary and retroauricular approach; RA, Retroauricular approach. * Statistically significant, *p*-value < 0.05.

**Table 3 jcm-11-03320-t003:** Operation times for the total thyroidectomy and modified neck dissection patients.

Patients	Total Operation Time (min)	Flap Elevation and Dissection under Direct Vision (min)	Robot Docking for RAND (min)	Console Time for RAND (min)	Flap Elevation for TT (min)	Robot Docking for iT (min)	Console Time for iT (min)	Robot Docking for cT (min)	Console Time for cT (min)
TA5	530	-	5	157	3	105			62
TA6	520	266	-	-	11	130		5	50
TA7	538	90	5			265			
RA10	310	135	5	110			10	5	50
RA11	430	185	5	40				5	63
RA12	310	25	10	250					
RA13	300	30	11	205					
RA14	358	15	12	314					
RA15	280	12	12	247				-	-
Con6	340								
Con7	360								
Con8	410								

Abbreviations: RAND, robot-assisted neck dissection; TT, total thyroidectomy; iT, ipsilateral thyroidectomy; cT, contralateral thyroidectomy.

## Data Availability

Not applicable.

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
