# Peer review of "Robot-Assisted Total Thyroidectomy with or without Robot-Assisted Neck Dissection in Pediatric Patients with Differentiated Thyroid Cancer"

_jcm, 2022, doi:10.3390/jcm11123320_

Round 1

Reviewer 1 Report

1. Title should be "...in pediatric patients with well-differentiated...". And throughout, there is no "pediatric thyroid cancer". It should be "pediatric patients with thyroid cancer" or "thyroid cancer in pediatric patients". 

2. Remove all abbreviations from abstract

3. Far too many abbreviations, most of which are not commonly known. You should remove nearly all abbreviations in this paper, except commonly used ones like CT, MRI. This paper is nearly unintelligible because of abuse of abbreviations. Be sure to remove TT, RAND, TARA, RA, TA, PTC, CCND, HT, SAN, MND and nearly all others from the text. Abbreviations are OK in tables as long as they are defined in the table. 

4. Abbreviations in Figures must be defined in the figure legend (e.g. Figure 1 has NO definitions.)

5. The title specifically mentions "well-differentiated thyroid cancer" and in the paper you talk about "differentiated thyroid cancer"....terminology should match.

6. Line 23 is the end of the discussion in the abstract which should be present tense.

7. You talk about pediatric patients and in methods you state <20yo but you never formally define the age of pediatric patients in your study.

8. Pediatric patients should not be referred to as women and men but are males and females (Line 82)

9. Line 89, remove "on the other hand". This is informal conversational jargon unsuitable for a scientific paper. (and anywhere else this phrase is used)

10. Line 149 "Harmonic" should not be capitalized [is this really the J&J trade name???]

11. Line 197 and 202 and 204  "Robotic" should not be capitalized in the middle of a sentence

12. Line 201 "subanalyze" is not appropriate English. You should discuss subgroup analysis

13. You keep describing surgery as "robotic" but in fact it is "robot-assisted". Change  throughout. For example "Robotic TT" should be "Robot-assisted TT" [but without the terrible abuse of abbreviations]

14. You talk about and document longer operating time with the robot-assisted approach but you make no mention (that I saw) of the differences in cost. You need to address this issue even if in general terms. Most readers will not know how longer operating time affects cost in the Korean health care system. Who pays for this?

15. Line 341 should be "metastases"

16. Line 324 How did the robot-assisted approach affect rate of lymphedema?

17. Line 343 It would seem inappropriate to refer to robot-assisted thyoid surgery as the "optimal" alternative based on the data in this pilot study. 

18. The authors use the term "precision" five times and I am not convinced that they present evidence regarding "precision". 

19. In Table 1, do not use the word "Case" to indicate "patients". Column 2 should be "Gender" and not "sex". 

20. Line 381 remove "Moreover" (and in all other places it is used)

21. Line 324 you say less lymphedema, but this word is not used anywhere else in the paper, so where do you document less lymphedema in the robot-assisted group?

22. You discuss comparison of results of robot-assisted and conventionally operated groups (e.g. line 229). You state that robot-assisted is feasible, which from your data I agree with. HOWEVER!!! I do not see any statements to suggest why the robot-assisted technique is BETTER, based on measured outcomes or data. If there is no objective "better" for the robot-assisted procedure then you must state that clearly. If there is objective evidence that robot-assisted is better in yur series, then state that clearly. 

23. Line 248 Why is "Parathyroid" capitalized"

24. Line 196 What is "approximately longer operation time".... this is NOT approximate!!! It is a statistically significantly longer time as you state in nthe next line. Your words would appear to be deliberately misleading. 

Overall this is an interesting report of robot-assisted thyroid surgery in pediatric patients. I found the paper extremely tedious and almost impossible to read because of the consistent abuse of abbreviations. In its current form, I cannot subject the readership to this. If the paper is revised appropriately, then it could become a valuable addition to the literature. There needs to be a clear statement about the robot-assisted approach if it is better or not based on  objective measures. While the authors state that it "can be an optimal alternative", I do not see any objective evidence to support this statement. If this evidence exists it needs to be clearly stated. 

I hope that the authors will choose to revise this paper and look forward to reviewing a readable version without the abbreviations. 

Reviewer 2 Report

Your  paper shows  a good casuistic  and   good presentation of reported cases . 

Tell us how  the patients choose  a conventional or a remote access.  

In my opinion this kind of operation ( remote access) for thyroid deseases  does not be called minimally invasive . The  flaps and dissections is quite bigger from conventional.  Please would you   disccus this point better?

The other point that you must improve is  a discussion of operation time . You 'd pointed  a statiscal study that shows  no  difference  between the aproaches ( conventional   and remote acces) . But  you can see that  the operation time is quite different in table 2.  I my opinion  the patients stayed under  general anestesia  mucho more  in the remot access. 

 You 'd showed no complications in  scar. Would you  please  disccuss the sensiblity of the skin   as the  procedure  to prepare  the flap may  damage  sensitive nerve of the cervical plexus ?

At the end  , your  study  does not   show  a  randomized  aspect .  Please    discuss this issue  in details.

Round 2

Reviewer 1 Report

The paper is much improved and I thank and congratulate the authors for doing a nice job. 

1. I apologize for not catching this earlier, but all percentages can only be reported to 2 significant figures. See lines 209, 358, 362 and maybe others.

2. Line 320 you say "cases". This word should NEVER be used to mean "patients". Check throughout and change to "patients".

The paper should be publishable with these minor changes and I sincerely thank the authors for their excellent contribution. 

Author Response

Thank you so much for the review.

I have changed the percentage marks, within 2 significant figures, except for the first line expressing ("Although thyroid cancer rarely occurs in childhood, it accounts for approximately 0.7% of all cancers")_ as this was a quotation from the reference that was marked.

Also, I have replaced the misnomer "cases" to "patients" throughout the manuscript, thanks for your suggestion.